# The Effect of Dicer Knockout on RNA Interference Using Various Dicer Substrate Small Interfering RNA (DsiRNA) Structures

**DOI:** 10.3390/genes13030436

**Published:** 2022-02-27

**Authors:** Min-Sun Song, Jessica Alluin, John J. Rossi

**Affiliations:** 1Center for RNA Biology and Therapeutics, Beckman Research Institute of City of Hope, Duarte, CA 91010, USA; msong@coh.org (M.-S.S.); jalluin@coh.org (J.A.); 2Irell and Manella Graduate School of Biological Sciences, Beckman Research Institute of City of Hope, Duarte, CA 91010, USA

**Keywords:** dicer, tetra-loop, dicer-substrate siRNA (DsiRNA), RNA interference (RNAi), RNA biogenesis, dicer knockout, CRISPR system

## Abstract

Small interfering RNAs (siRNAs) are artificial molecules used to silence genes of interest through the RNA interference (RNAi) pathway, mediated by the endoribonuclease Dicer. Dicer-substrate small interfering RNAs (DsiRNAs) are an alternative to conventional 21-mer siRNAs, with an increased effectiveness of up to 100-fold compared to traditional 21-mer designs. DsiRNAs have a novel asymmetric design that allows them to be processed by Dicer into the desired conventional siRNAs. DsiRNAs are a useful tool for sequence-specific gene silencing, but the molecular mechanism underlying their increased efficacy is not precisely understood. In this study, to gain a deeper understanding of Dicer function in DsiRNAs, we designed nicked DsiRNAs with and without tetra-loops to target a specific mRNA sequence, established a Dicer knockout in the HCT116 cell line, and analyzed the efficacy of various DsiRNAs on RNAi-mediated gene silencing activity. The gene silencing activity of all DsiRNAs was reduced in Dicer knockout cells. We demonstrated that tetra-looped DsiRNAs exhibited increased efficacy for gene silencing, which was mediated by Dicer protein. Thus, this study improves our understanding of Dicer function, a key component of RNAi silencing, which will inform RNAi research and applications.

## 1. Introduction

Small interfering RNAs (siRNAs) are artificial molecules used to silence genes of interest through the RNA interference (RNAi) pathway, mediated by the endoribonuclease Dicer. In fact, siRNAs efficiently induce sequence-specific gene silencing, which could affect the therapeutic potential of RNAi for disease-causing genes and cancer [1]. Many studies have shown the therapeutic tool of RNAi in various clinical trials. The world’s first RNAi drug of ONPATTRO (Patisiran) against transthyretin, in the treatment of hereditary transthyretin amyloidosis [2], led to regulatory approval by the FDA in 2018. Dicer-substrate small interfering RNAs (DsiRNAs) are an alternative to conventional 21-mer siRNAs, with a reportedly higher efficiency. DsiRNAs are processed by the enzyme DICER into the desired conventional siRNAs [3]. DsiRNAs are a useful tool for sequence-specific gene silencing, but the molecular mechanism underlying their increased efficacy is not precisely understood.

A hairpin structure is one of the most abundant RNA secondary structural elements. RNA hairpins play essential structural and functional roles by providing sites for RNA tertiary contacts and protein binding, which facilitates the assembly of ribonucleoprotein particles. More than half of all known RNA hairpins are composed of four nucleotides, called tetra-loops [4,5,6,7,8]. Consequently, RNA tetra-loops are predominant and are commonly found in nature. Tetra-loops have been found in many RNAs, including ribosomal RNA, mRNA, a group I intron ribozyme, RNase III, coxsackievirus B3, and heron Hepatitis B virus [9,10,11,12,13]. In general, RNA tetra-loops are more stable compared to smaller or larger loops accommodating the same stem because the compact structures of tetra-loops bestow high thermal stability and nuclease resistance. These loops involve base stacking, base-phosphate, and base-ribose hydrogen bonds [14]. GNRA and UNCG (5′ to 3′, N: any base, and R: purine) are the most common among the known stable tetra-loops [15,16]. Among these families, the GAAA tetra-loop has a U-turn motif, which can be involved in RNA activity by facilitating both RNA–RNA and RNA–protein interactions [17]. It is noteworthy that RNA tetra-loops confer functional roles within the RNA beyond allowing secondary structure formation. The tetra-loops generally participate in RNA tertiary interactions with other RNAs and RNA–protein interactions [14,18]. Whether tetra-loops affect processing by Dicer is currently unknown.

Previously, chemically synthesized 25- to 27-nucleotide (nt)-long double-stranded RNAs with 2-nt 3′ overhangs were identified as Dicer substrates. These Dicer-substrate siRNAs (DsiRNAs) are recognized and processed into shorter small interfering 21–22 bp RNAs (siRNAs) by endogenous Dicer when they are introduced into mammalian cells [19]. The interaction between DsiRNAs and Dicer promotes the loading of siRNAs into the RNA-induced silencing complex (RISC) with strand-specific orientation and the association of the guide strand with the argonaute protein Ago2 in the cytoplasm [19,20,21]. Consequently, DsiRNAs have been reported to be up to 10-fold more potent in silencing a targeted gene than canonical 21-nt siRNAs [22,23,24]. In contrast, endogenous microRNA (miRNA) is processed from hairpin-containing primary transcripts (pri-miRNA) of nuclear hairpin double-strand RNA into pre-miRNA by the ribonuclease Drosha, before it is transported to the cytoplasm and further cleaved by Dicer [25]. RNA containing hairpin structures influences Dicer’s activity and site selection [26]. The loop structure of short hairpin RNAs (shRNAs), which are artificially designed small RNAs, contributes to the biogenesis and subsequent activity of siRNA with Dicer [27]. An important question is how the activity of small RNAs is improved by interaction with Dicer. The answer may lie in studying the structures between small RNAs and Dicer. A cryo-electron microscopy structural study showed that the terminal loop of pre-miRNA interacts with the N-terminal DEAD-like helicases domain (DExD)/H-box helicase domain of Dicer and its cofactor, transactivation response element RNA-binding protein (TRBP) [28]. This implies that the presence of loops within small RNAs may influence not only Dicer cleavage activity but also gene silencing efficacy.

At present, it is not clear how tetra-loops affect Dicer activity or whether Dicer can directly affect the RNAi activity of DsiRNAs. To clarify this, in this study, we designed various DsiRNAs with or without a 5′-GAAA-3′ tetra-loop and stem structure and determined whether the various versions of DsiRNAs affected Dicer-dependent gene silencing efficacy. To confirm that Dicer is required for processing DsiRNAs, we generated Dicer knockout cells (named H2-2) in the colorectal cancer cell line HCT116. We compared the gene silencing activity of the various DsiRNAs in wild-type (WT) HCT116 and Dicer-inactivated H2-2 cells. Compared to WT cells, the gene silencing activity of DsiRNAs, including both original and tetra-looped versions, was reduced 2- to 37-fold in H2-2 cells, indicating a requirement for Dicer. These results have important ramifications for the role of Dicer in the efficacy of DsiRNAs and tetra-looped DsiRNA biogenesis.

## 2. Materials and Methods

### 2.1. DsiRNA Duplex

All RNA strands for DsiRNAs were synthesized as single-strand RNAs (ssRNAs) by Integrated DNA Technologies with high-performance liquid chromatography purification and resuspended in RNase-free water (Table 1). ssRNAs were annealed to form DsiRNA duplexes at 95 °C for 5 min, then incubated for 4 h at room temperature, before being aliquoted (10 µL in a 1.5 mL tube) and stored at 80 °C.

### 2.2. Cell Culture and Transfection

HCT116 cells were cultured in DMEM supplemented with 10% fetal bovine serum and penicillin/streptomycin at 37 °C in 5% CO2 with humidification. HCT116 cells were transfected with Lipofectamine 2000 (Life Technologies, Carlsbad, CA, USA). The Dicer knockout HCT116 cells were generated by employing CRISPR technology (DICER Double Nickase Plasmid h2, Santa Cruz, sc-400365-NIC-2) following the manufacturer’s protocol. The DICER genome sequence and sgRNA targets are represented in Figure 1. Single-cell clones (H2-1, H2-2, and H2-3) were selected five to six weeks following their inoculation.

### 2.3. Surveyor Nuclease Assay

We purified gDNA from CRISPR/Cas9-treated HCT116 cells to confirm Dicer knockout. Genomic DNA was extracted using the purelink genomic DNA mini kit (Invitrogen Carlsband, USA) following the manufacturer’s protocol. We amplified segments containing the sgRNA target site by PCR, using 100 ng gDNA with a representative primer (Figure 1B; forward 5′-CTGTTTGAAGGGTAGGA-3′ and reverse 5′-GCCTGAAAGGGTAAAATG-3′). A 663 bp amplicon from the control templates was PCR amplified from 100 ng of control C and control G (Surveyor^®^ mutation detection kit, IDT, San Diego, CA, USA). The sequence of the PCR product for control G is shown in Appendix A. Control C has a cytosine base in lieu of the guanine base (red G in Appendix A). The target regions for control G and C were amplified with the forward (5′- ACACCTGATCAAGCCTGTTCATTTGATTAC-3′) and reverse (5′- CGCCAAAGAATGATCTGCGGAGCTT-3′) primers. We purified the DNA from PCR products (Appendix A) using the PCR-clean up kit (Macherey-Nagel, Duren, Germany), then mixed 300 ng PCR products obtained from WT/Dicer knockout mutant HCT116 cells and control G/control C. We denatured them by heating at 99 °C for 5 min in a thermocycler. We then formed heteroduplexes and homoduplexes by cooling to room temperature. 

We performed the Surveyor nuclease assay using a Surveyor^®^ mutation detection kit (IDT). We mixed each sample with 1 µL of Surveyor Enhancer S, 1 µL of Surveyor Nuclease S, and 4 µL of 0.15 M MgCl_2_ in a 50 µL reaction. We incubated the mix for 60 min at 42 °C and stopped the reaction by adding 4 µL of the stop solution provided in the kit. The reactions were either kept at −20 °C or used immediately for electrophoresis.

### 2.4. Sequencing Analysis

For sequencing analysis at the Dicer locus, the genomic region including the target site of sgRNAs was amplified with the primers (Figure 1B; forward 5′-CTGTTTGAAGGGTAGGA-3′ and reverse 5′-GCCTGAAAGGGTAAAATG-3′). A 1158-bp amplicon of gDNA from HCT116 and H2-2 clone were subcloned into pCRBlunt-TOPO (Invitrogen, Carlsband, USA). The amplicons containing the cloned genomic sequence (negative control, H2-2_1 and H2-2_2) was amplified from each colony using the M13 forward and reverse primers prior to Sanger sequencing (Figure 2B).

The Dicer mRNA in which the CRISPR editing area from sgRNA was amplified with forward (5′-AGAAACACTGGATGAATGA-3′) and reverse (5′-CAAAGAAAGGACCCATTG-3′) primers. We amplified the segments containing the Dicer mRNA (Figure 3C). The 99-bp from HCT116 and 86-bp from H2-2 amplicons were subcloned into the pCRBlunt-TOPO (Invitrogen) and sequenced using M13 forward and reverse primers.

### 2.5. Dual-Luciferase Assay

To generate the reporter plasmids psi-hnRNPH-S (sense reporter) and Psi-hnRNPH-AS (antisense reporter), a 343-bp PCR fragment of hnRNPH cDNA (Acc.: NM_005520) was cloned in the 3′-UTR of the humanized Renilla luciferase gene in the psiCHECKTM-2 vector (Promega, Madison, WI, USA) in either the sense or antisense orientation. The cells were co-transfected with 0.1 µg of the plasmid with the dual-luciferase reporter system and DsiRNAs using 2 µL of Lipofectamine 2000 in a 48-well plate. Luciferase assays were performed 48 h after transfection, using the dual-luciferase reporter assay system (Promega, Madison, WI, USA). Firefly luciferase activity was normalized to Renilla luciferase activity and then to its own control, the activity of which was set to 100.

### 2.6. Statistical Analysis

Statistical analyses were performed using a Student’s *t*-test. All data represent the mean ± standard deviation (S.D.) of at least three independent experiments. Student’s *t*-test ** p* < 0.05, *** p* < 0.01, **** p* < 0.001, and ***** p* < 0.0001

## 3. Results

### 3.1. Design of DsiRNAs Targeting hnRNPH1

We previously observed successful gene silencing activity using DsiRNAs that targeted the RNA-binding protein heterogeneous nuclear ribonucleoprotein H1 (hnRNPH1); this activity was associated with Ago2, TRBP, and Dicer [20,29]. In the present study, we synthesized DsiRNAs specific to hnRNPH1 that also contained various tetra-loop and stem structures (Figure 4; Table 1). DsiRNAs to enhance the efficiency of Dicer-mediated loading of siRNA into the RISC were designed as asymmetric duplexes containing a 27-base antisense strand with a 2-nt 3′-overhang and a 25-nt sense strand. Two DNA nucleotides were included at the 3′ end of the sense strand to create a blunt end 3. These asymmetric 25/27-mer siRNAs were optimized for processing by Dicer. We designed two versions of the DsiRNA (DsiRNA I [DI] and DsiRNA II [DII]), which differed by only a single base pair. Previously, our results showed that hnRNPH1-targeted DsiRNAs could show strand selectivity; DsiRNA I has similar selectivity for either strand, but DsiRNA II has more substantial activity on the antisense strand 20. We added a 5′-GAAA-3′ tetra-loop (TL) into each DsiRNA and synthesized different stem structures for each, one with a GC-rich stem (TL_DI and TL_DII), and another with the original stem (TL_DI_O and TL_DII_O), which corresponds to the hnRNPH1 target gene. In previous studies, siRNA containing a nick, called small internally segmented interfering RNA (sisiRNA), was capable of effectively knocking down endogenous gene expression to the level of canonical siRNA while abolishing the off-target silencing effect induced by complementary guide strands [30]. For this reason, we designed a nick in the guide strand of DsiRNAs, which we also expected to improve the efficiency of DsiRNAs (Figure 4).

### 3.2. Generation of Dicer Knockout HCT116 Cells

We previously showed that chemically synthesized 25- to 27-nt-long DsiRNAs interact with Dicer to facilitate the loading of small RNAs into RISC [3]. DsiRNAs increase the gene silencing effect up to 100 times more efficiently than canonical siRNAs [20]; recruiting the Dicer enzyme complex using DsiRNAs improves RISC assembly and gene silencing efficacy compared to 21-mer siRNAs. To investigate the effect of tetra-looped DsiRNAs on Dicer efficacy, we generated Dicer knockout cells by transfecting RNA-guided Cas9 endonuclease into the HCT116 cell line. We chose the HCT116 cell line because it harbors a single copy of each chromosome, thus reducing the challenges frequently associated with achieving homozygosity in diploid cells for genetic studies [31]. Based on karyotyping, its stemline chromosome number is near diploid, with the modal number at 45 (62%) and polyploids occurring at 6.8% [32]. To knock out DICER in the human cell line HCT116, we used CRISPR/Cas9 technology. We designed guide RNAs complementary to the area near the genomic locus corresponding to the DExDc of DICER (Figure 1A,B; blue dot and letters). We included an indicator construct containing a green fluorescent protein (GFP) to show specific GFP expression in cells expressing Cas9. To generate a Dicer-deficient derivative of HCT116 cells, we isolated and expanded single-cell clones for further analysis. We obtained a series of three independent clonal cell lines (H2-1, H2-2, H2-3). After single-cell cloning, we purified genomic DNA (gDNA) and performed PCR using gDNA primers (Figure 1B, represented by green or red arrows; Appendix A). We mixed WT PCR product from parental HCT116 cells with Dicer H2 PCR product from single-cell clones of Dicer knockout HCT116 cells in equal quantities to produce heteroduplex molecules, then performed Surveyor nuclease assays and analyzed the cleavage products using DNA gel electrophoresis. By design, we expected the Surveyor nuclease enzyme to cleave the mismatched nucleotides in the heteroduplex molecule and generate two bands of 962 bp and 196 bp, whereas the homodimer PCR products would show an undigested fragment. The Dicer H2-2 clone showed the expected two bands in our electrophoresis assay (Figure 5). This indicated that Dicer H2-2 contained the mismatched nucleotides introduced by the CRISPR/Cas9 system, and it was selected for further analysis.

To confirm Dicer knockout, genomic DNA was purified from WT HCT116 and CRISPR/Cas9-treated Dicer knockout HCT116 cells and amplified using PCR; products were mixed, then denatured and allowed to form heteroduplexes and homoduplexes, then subjected to Surveyor nuclease assay and separation by gel electrophoresis. The positive control was the control C/G, a 633 bp control DNA with a point mutation (Appendix A), demonstrating the expected 416 bp and 217 bp bands. Homoduplex DNA without mismatch did not cleave the nuclease, but heteroduplex DNA (Dicer H2-1, H2-2, and H2-3) shows a cleavage band in generating site-specific double-strand breaks (Figure 5). Hereafter, all clones tested were efficacious in facilitating the cleavage of DNA at specific targets in the Dicer genome.

We analyzed the gDNA using Sanger sequencing to confirm that the mismatched nucleotides were included in the Dicer H2-2 clone. Using the CRISPR/Cas9 system, we designed two single-guide RNA (sgRNA; 20 bp: blue letters in Figure 2A) followed by a PAM sequence (orange letters in Figure 2A). We sub-cloned the PCR products and then sequenced each clone. All three clones had altered sequences (Figure 2B). Subsequent sequence analysis confirmed that the CRISPR/Cas9 system introduced DNA double-strand breaks at the target genomic sequences and thereby induced indels via error-prone nonhomologous end-joining repair [33].

We performed western blot analysis with anti-Dicer antibodies to confirm Dicer protein expression in HCT116 and its absence in Dicer knockout H2-2 cells (Figure 3A). In H2-2 cells, we observed the anticipated loss of protein, presumably due to the mutation in Dicer exon 2; this loss was detected by the anti-Dicer antibody Ab14601, which only showed a Dicer band in the parental HCT116 cells. However, the anti-Dicer antibody Ab13502 detected the appearance of a novel non-specific band in H2-2, which was the same size as the Dicer protein. To confirm that Dicer was knocked out in H2-2 cells, we compared the amino acid sequence in gDNA isolated from HCT116 and H2-2 cells (Figure 3B). Figure 3C shows the Dicer mRNA sequence targeted with dicer gRNA. The H2-2 Dicer knockout cells exhibited a deletion of three amino acids and a mutation of seven amino acids. The 9-nt missing still in-frame in Dicer proteins and the potential AUG initiation codons in the H2-2 genomic DNA sequencing can generate open reading frames that overlap the Dicer reading frame (Appendix A). To overcome the potential shortcomings of this approach, which include, for example, initiation of translation from an alternative Dicer expression that is hard to quantify, we analyzed the impact of Dicer expression via mass spectrometry. Dicer unique peptides only showed in the sample from HCT116, which generated the gel band extraction from closed to Dicer protein size in Appendix A. We used the TaqMan^TM^ MicroRNA assay to establish the temporal dynamics of miRNA depletion following induction of Dicer loss of function (Appendix A). miR-1254 is a non-canonical miRNA produced from an intron of protein-coding gene, CCAR1. The position of pre-miR-1254 overlaps with that of the Alu sequence, a type of short interspersed nuclear element (SINE), and belongs to the Alu Jr subgroup. miR-1254 is dependent strictly on Dicer [31]. miR31 is a canonical miRNA. Our analysis of the abundant miR-31 and miR-1254 in HCT116 and H2-2 confirmed that the levels of the abundant miRNAs in Dicer knockout cells are at least 90% (miR-31) and 70% (miR-2154) lower than in wild-type cells (Appendix A). To confirm Dicer knockout by measuring the expression of Dicer mRNA, we generated a Dicer primer targeting the mutation area and performed RT-PCR. We identified a slightly smaller band amplified from H2-2 cells compared with the band amplified from HCT116 cells (Figure 3D). We generated a DNA sequence from the PCR bands to identify the band. DNA from two colonies of HCT116 cells exactly matched the expected Dicer mRNA sequence size of 99 bp (Appendix A), but the DNA sequences from four colonies of H2-2 cells did not align with the Dicer mRNA sequence (89 bp: Appendix A). To sanction the source of the inserted sequence, we used NCBI nucleotide blast. The results showed that the band amplified from Dicer knockout H2-2 cells is a PCR product from human transformer two α homolog (TRA2A) mRNA. Thirteen nucleotides at the 3′ end of the forward Dicer primer and 11 nucleotides at the 3′ end of the reverse Dicer primer perfectly matched the TRA2A mRNA. These findings show that neither Dicer protein nor Dicer mRNA are expressed in H2-2 cells.

### 3.3. Tetra-Looped DsiRNA Enhances siRNA Efficacy

To determine the efficacy of tetra-looped DsiRNA, we used dual-luciferase assays in WT HCT116 cells to separately detect the level of gene silencing conferred by the antisense or sense strand. We normalized the relative firefly luciferase activity of the DsiRNAs by dividing by Renilla luciferase as an internal control, then converting activity to 0–100% of control. We first confirmed the suppression efficiency of DsiRNA in WT HCT116 cells. The DsiRNA I sense strand showed significantly greater suppression activity than the antisense strand (Figure 6A); in contrast, the DsiRNA II antisense strand showed significantly greater suppression activity than the DsiRNA II sense strand (Figure 6B). The inclusion of tetra-looped DsiRNAs did not change overall strand selectivity in WT HCT116 cells, except with TL-DsiRNA II (Appendix A). Next, we evaluated the degree of the silencing effect of each strand of tetra-looped DsiRNAs in WT HCT116 cells. The gene silencing potency of tetra-looped DsiRNA significantly increased on the preferred sense strand for DsiRNA I (Figure 6C) and the preferred antisense strand for DsiRNA II (Figure 6F). However, RNAi activity was inconsistent on the passenger strand of DsiRNA I_AS and DsiRNA II_S (Figure 6D,E). We designed a nick break between nucleotide bonds that we expected to be cut by Dicer and added tetra-loops to the DsiRNA. On the other hand, an intact guide strand that is selected by Ago2 is not necessary for cleavage by Dicer. The resulting increase in gene silencing activity of tetra-looped DsiRNAs might easily unwind from RISC as a nick break on the preferred strand. These findings suggest that the tetra-looped DsiRNAs are more efficient for gene silencing than the original DsiRNAs.

### 3.4. Gene Silencing Activity is Controlled by Dicer

To confirm that Dicer is the main protein involved in small RNA biogenesis, we transfected the DsiRNAs and tetra-looped DsiRNAs into WT HCT116 cells and H2-2 Dicer knockout HCT116 cells, then conducted luciferase assays to quantitate gene silencing. If DsiRNA-induced gene silencing does not occur through Dicer, we would expect equivalent levels of luciferase activity in both WT HCT116 and H2-2 Dicer knockout cells. After Dicer knockout, all DsiRNAs showed good gene-silencing activity (Figure 7). These results support that there exist noncanonical pathways for DsiRNA biogenesis, which bypass a part of the biogenesis steps of Dicer cleavage. As for Dicer-independent biogenesis, mature miR-451 can be produced without the Dicer cleavage step. Pre-miR-451 is cleaved by Ago2 in the middle of the 3′ strand, and further trimmed by 3′-5′ exoribonuclease PARN to yield a mature form of miR-451 [34]. DsiRNAs may have alternative pathways, such as pre-miR-451, that do not require a Dicer cleavage step during biogenesis. However, all strands of DsiRNAs, including tetra-looped DsiRNAs, showed significantly reduced gene silencing activity in H2-2 Dicer knockout cells compared to WT HCT116 cells (Figure 7). These results show that Dicer processing is essential to small RNA-mediated gene silencing of both DsiRNAs and tetra-looped DsiRNAs. 

## 4. Discussion

In this study, we evaluated the gene silencing activity of DsiRNAs and tetra-looped DsiRNAs in WT and H2-2 HCT116 cells to better understand Dicer mechanisms. We found that tetra-looped DsiRNA improved RNAi-mediated gene silencing activity (Figure 6). Previously, shRNA, which has a stem-loop structure, was shown to induce stable and long-lasting gene silencing activity [35,36,37]. The RNAi gene silencing activity of shRNAs with stem-loop structures requires Dicer activity [26], possibly because the terminal loops of shRNA and pre-miRNAs share similar RNA structures, which interact with the N-terminal helicase domain of Dicer [28,38,39,40]. Introducing a nick into siRNAs (i.e., sisiRNAs) is an effective strategy to increase the specificity of the siRNA, allowing greater specificity for biological action [30,41]. Our results suggest that tetra-loop structures are involved in essential Dicer ribonuclease functions, and may have potential alternative effects through interaction with RNA and proteins. Crystal structure analyses of Dicer and pre-miRNAs revealed that the N-terminal domain of Dicer binds the more stable region of single-stranded hairpin looped pre-miRNA [42,43,44]. The N-terminal domain of Dicer is bound to TRBP, which advances the RNA-binding affinity of Dicer [45] and reinforces cleavage accuracy [42,46]. We suggest that Dicer possesses not only ribonuclease activity but can also affect RNA stability and that TRBP is an RNA cofactor for Dicer complexes, such as for DsiRNA with tetra-loops, which can increase gene silencing activity. This suggestion agrees with previous reports indicating that Dicer can globally bind to stem-looped RNAs without cleavage activity and influences the fate of targeted transcripts by gene silencing [47]. Our experimental data showed that DsiRNAs with added tetra-loops showed higher efficiency in gene silencing, but this activity disappeared in Dicer knockout HCT116 cells (Figure 6; Figure 7). This finding suggests a potential complex in which the stem-loop structures of small RNAs interact with Dicer to improve gene silencing activity.

As far as Dicer knockout HCT116 cells are concerned, although Dicer helicase mutant HCT116 cells have been used to validate the efficiency of various shRNAs in our previous studies [48]. In previous studies, Dicer helicase mutant HCT116 did not affect the mature short RNA generation of miR-23a, miR-27a, and artificial shRNAs. This has proved to be very difficult because of Dicer helicase mutant HCT116 with defects in the processing of most, but not all, endogenous pre-miRNAs into mature miRNA. Another college group showed that a stem-looped miRNA does not require a Dicer cleavage step during its biogenesis [34,49]. The biogenesis of miR-451 occurs independently of Dicer and instead requires cleavage of the 3′ arm of the pre-miR-451 precursor hairpin by Ago2. This is evidence that not all stem-looped RNAs are affected by their efficiency by Dicer. Our studies demonstrated how we improved the knockout cells and knockout validation by the CRISPR system. To obtain the Dicer knockout cell lines, we consider three things. First, we chose the HCT116 cell line because it is near diploid and is often used for gene knockout studies [31,32]. The diploid cell lines offer a complete loss-of-function phenotype from a single allele knockout and eliminate any activity of the knockout from a second allele seen in diploid cell models [50,51,52]. Second, we collected a single cell harboring Dicer knockout after transfection of CRISPR plasmid selected GFP-only expression cells by FACS sorting. The method can generate a new cell line that can be verified as complete knockouts [53,54,55]. Finally, we used two methods of western blot and Dicer mRNA from qRT-PCR to confirm the knockout validation. The Dicer antibody of Ab13502 detected the band from lysates of H2-2, which was Dicer knockout cells. However, the Dicer Ab14601 antibody exhibited target specificity and sensitivity to allow the identification of the Dicer protein (Figure 3A). It is of utmost importance to find an excellent antibody to reduce cross-reactivities with off-target proteins that can lead to the recognized issue of experimental irreproducibility [56,57]. We clearly confirmed the knockout validation from the sequences detected of Dicer mRNA from HCT116 and H2-2 (Figure 3D and Appendix A). 

CRISPR/Cas9 technology enables the rapid generation of loss-of-function mutations in a targeted gene in mammalian cells. A single cell harboring those mutations can be used to establish a new cell line, thereby creating a CRISPR-induced knockout clone. After introducing the Dicer gRNAs and Cas9 expressed plasmid via transfection into HCT116, single cells must be isolated in order to generate clonal lines that can be verified as complete knockouts. We isolated single cells by fluorescence-activated cell sorting (FACS) and plated them into 96-well plates by diluting 0.8 cells per well. Cell clones were expanded by plating to progressively larger plates until there were enough cells to freeze and take DNA and protein lysates for knockout verification. At this step, we got only three cell clones, which we named H2-1, H2-2, and H2-3 (Figure 5). We found that the growth time of H2-1 and H2-3 cells was prolonged in comparison to H2-2 in the cell expansion step. Finally, we selected only H2-2 for taking genomic DNA and protein lysate in order to confirm Dicer knockout. The Dicer knockout cells displayed only one cell clone in this experiment, presumably due to a deficit in miRNA biogenesis.

Dicer is an essential protein for small RNA biogenesis and has recently been reported to act as a multifunctional protein; its activity is not limited to miRNA and siRNA biogenesis [58]. The processing of long transfer RNAs (tRNAs) to small RNAs, broadly termed tRNA fragments (tRFs), is dependent on Dicer [59,60,61]. Sno-derived RNAs (sdRNAs), which are derived from small nucleolar RNAs (snoRNAs), showed reduced expression in Dicer mutants [62,63]. The depletion of most miRNA species is detected following Dicer ablation [31]. Our study provides evidence that tetra-loop DsiRNAs exhibit more potent gene silencing that depends on Dicer (Figure 7). We thus speculate that Dicer is not only identified by its endonuclease activity for small RNAs but also can be stably bound with tetra-looped DsiRNA to enhance gene silencing activity. 

Our observations raise intriguing questions regarding the mechanism of how Dicer improves the activity of RNA gene silencing. Interestingly, Dicer can bind to many classes of RNA molecules, in which the interaction does not necessarily lead to dicing. Specific stem-loop structures that bind with Dicer have been identified by human transcriptome-wide analysis and are called “passive Dicer-binding sites” [47]. Dicer is believed to participate in the correct guide strand for Ago2 loading to generate RISC [64,65]. Our findings suggest that Dicer can generate its own siRNA and can function to stabilize small RNA. We conclude that Dicer-mediated processing of tetra-looped DsiRNAs subsequently facilitates a more stable interaction and improved efficiency. 

## 5. Conclusions

DsiRNA has been reported to enhance the efficiency of Dicer-mediated loading of siRNA into the RISC. Tetra-looped hairpin of GAAA has been found to be more stable and nuclease resistant by their compact structure and high thermal stability. Notably, the Dicer activity of RNAi gene silencing has not yet been clearly demonstrated in RNAi research. Our present study included the following highlights: (1) Dicer knockout cells were developed using the CRISPR/Cas9 system; (2) The efficacy of tetra-looped DsiRNA was compared with linear DsiRNAs in colorectal carcinoma cell lines; (3) The gene silencing activity of various DsiRNAs disappeared in Dicer knockout cells; (4) The nicked tetra-looped DsiRNA required functional Dicer protein for silencing activity. Our research provides new insights into the mechanisms of Dicer underlying the gene silencing efficacy of various DsiRNAs.

## Figures and Tables

**Figure 1 genes-13-00436-f001:**
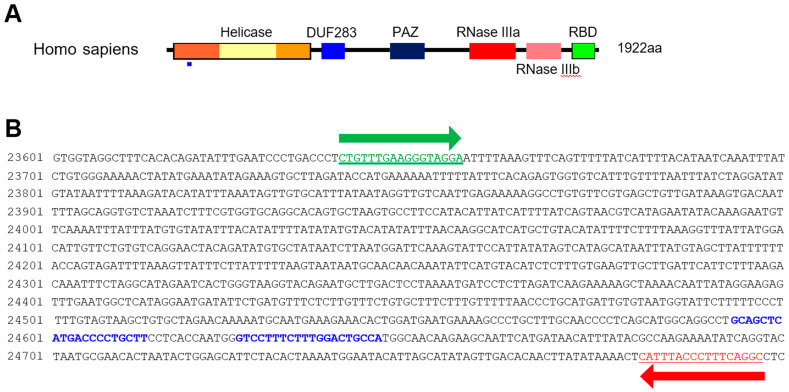
Dicer mutation using the CRISPR/Cas9 system. (**A**) Domain structure of Dicer protein. The small blue rectangle underneath indicates the region corresponding to the gDNA sequence targeted by the CRISPR/Cas9 system. (**B**) Sequence of DICER gDNA. The two sites with blue letters show the sequence targeted by double-nickase CRISPR. Green and red arrows indicate primers for detecting DICER gDNA.

**Figure 2 genes-13-00436-f002:**
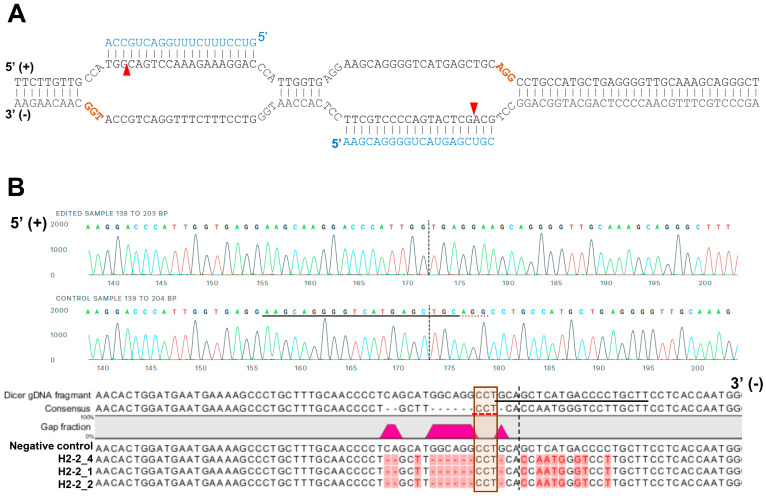
A representative sequence of the human DICER locus targeted by Cas9. (**A**) Schematic illustrating DNA double-nick approach using a pair of sgRNAs to guide Cas9 nickase. Cas9 can cleave only the strand complementary to the sgRNA (blue letters). A pair of sgRNA-Cas9 can cleave Dicer gDNA. Expected Cas9 cleavage sites are marked as red arrows. sgRNA offset is characterized as the distance between the PAM sequence (brown letters) and the 5′-ends of the guide sequence of a given sgRNA pair. (**B**) Representative sequences of the gDNA in the helicase domain of human DICER targeted by sgRNA. The PAM area is represented by a dotted red underline (upper) or red box (bottom). The dotted vertical line is the expected cleavage site. H2-2 clone sequences were compared with the human DICER gDNA sequence (XP_016876610:NCBI reference sequence). The negative control is the gDNA sequence from parental WT HCT116.

**Figure 3 genes-13-00436-f003:**
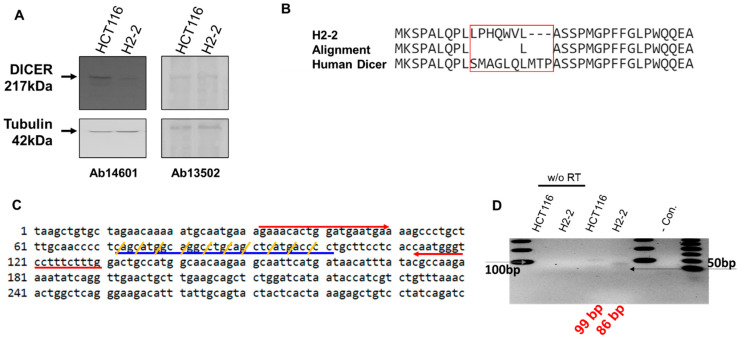
Analysis of Dicer protein and mRNA in HCT116 and Dicer knockout H2-2 cells. (**A**) Western blot experiments to confirm gene disruption in H2-2 Dicer knockout cells compared to WT HCT116 cells. The Dicer protein is 217 kDa in humans. Two anti-Dicer antibodies (Ab14601 and Ab13502) were used to detect the Dicer protein. Tubulin was used as a loading control. (**B**) Amino acid sequence analysis of the mutated area from H2-2 and human Dicer. The CRISPR/Cas9 system made nine nucleotide deletions and twenty-one nucleotide substitutions (Appendix A). The Dicer sequence from H2-2 cells shows a different protein-peptide compared with human Dicer (NP_001182502.1: NCBI reference number). (**C**) Human endoribonuclease Dicer mRNA sequence (1 to 300 nt: NM_001195573.1: NCBI reference sequence). Red arrows show the forward and reverse primers used to detect Dicer mRNA by PCR. The CRISPR/Cas9 system mutated the sequence indicated with blue underline. Yellow slashes indicate the codon of amino acid. (**D**) RNA was extracted from HCT116 and H2-2 cells and subjected to reverse transcription either with or without (w/o RT) the RT enzyme. cDNA was amplified by PCR with a Dicer-targeted primer (red arrow in (**C**)). The gray arrow is the DNA band of 100 bp in size. The last lane in the gel lane loads the low molecular weight DNA ladder (New England BioLabs Inc.: catalog number N3233). Black arrow is size of 50 bp.

**Figure 4 genes-13-00436-f004:**
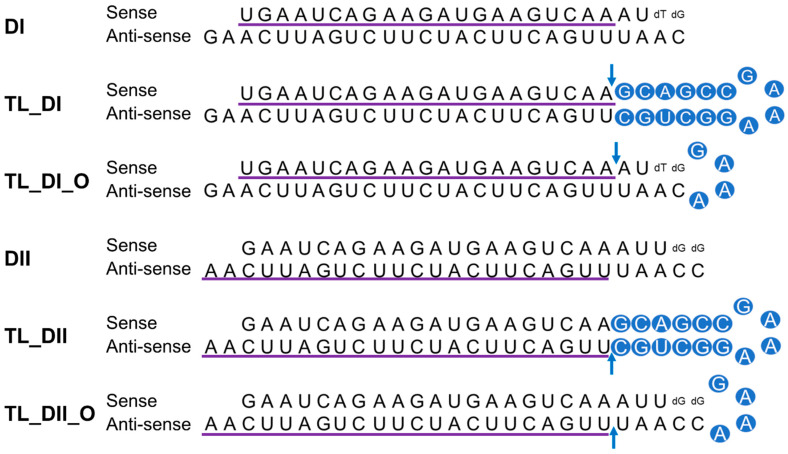
Structure of hnRNPH1-targeted DsiRNAs and tetra-looped DsiRNAs. We designed two DsiRNAs targeting hnRNPH1, DsiRNA I (DI) and DsiRNA II (DII), then added a 5′-GAAA-3′ tetra-loop (TL) into each DsiRNA and synthesized different stem structures for each, one with a GC-rich stem (TL_DI and TL_DII) and another with the original stem (TL_DI_O and TL_DII_O), which corresponds to the hnRNPH1 target gene. TL_DI has a 21-mer sense strand and a 38-mer antisense strand, with a GC-rich stem and 5′-GAAA-3′ tetra-loop (represented with blue circles). TL_DI_O has the same sequence as DI, with a 5′-GAAA-3′ tetra-loop sequence (blue circles). TL_DII has a 22-mer antisense strand and 36-mer sense strand, with a GC-rich stem and 5′-GAAA-3′ tetra-loop (blue circles). TL_DII_O has the same sequence as DII, with a 5′-GAAA-3′ tetra-loop sequence (blue circles). The detailed sequences are listed in Table 1. Ribonucleotides are shown in upper case and deoxyribonucleotides as dN. The purple underlined sequence shows the active strand. The blue arrow indicates the nick between ribonucleotides.

**Figure 5 genes-13-00436-f005:**
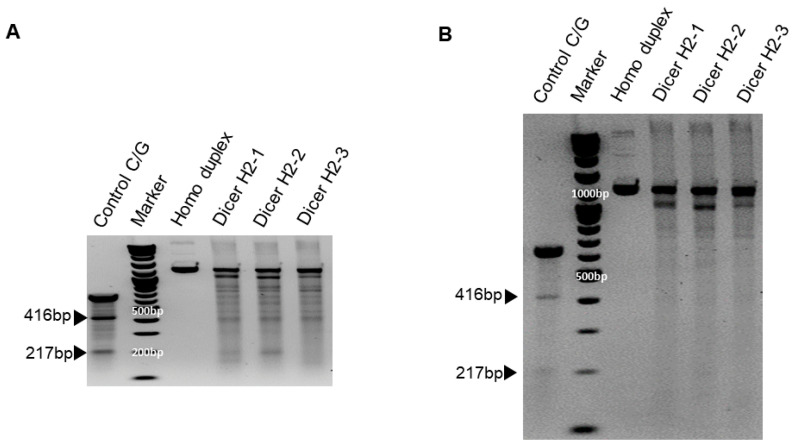
Surveyor assay comparing the efficiency of Cas9-mediated cleavage by double-nickase sgRNA in the human DICER locus. DNA duplex formation and treatment with a nuclease. SURVEYOR assay gel showing a comparable modification of control G/C, which is 633 base pair (bp) Control DNA with a point mutation (Appendix A) bearing 416 bp and 217 bp. Homoduplex without mismatch did not cleave the nuclease, but heteroduplex (Dicer H2-1, H2-2, and H2-3) shows the cleavage band. We ran the gel with short (**A**) and long (**B**) running times. Arrowheads indicate cleavage products.

**Figure 6 genes-13-00436-f006:**
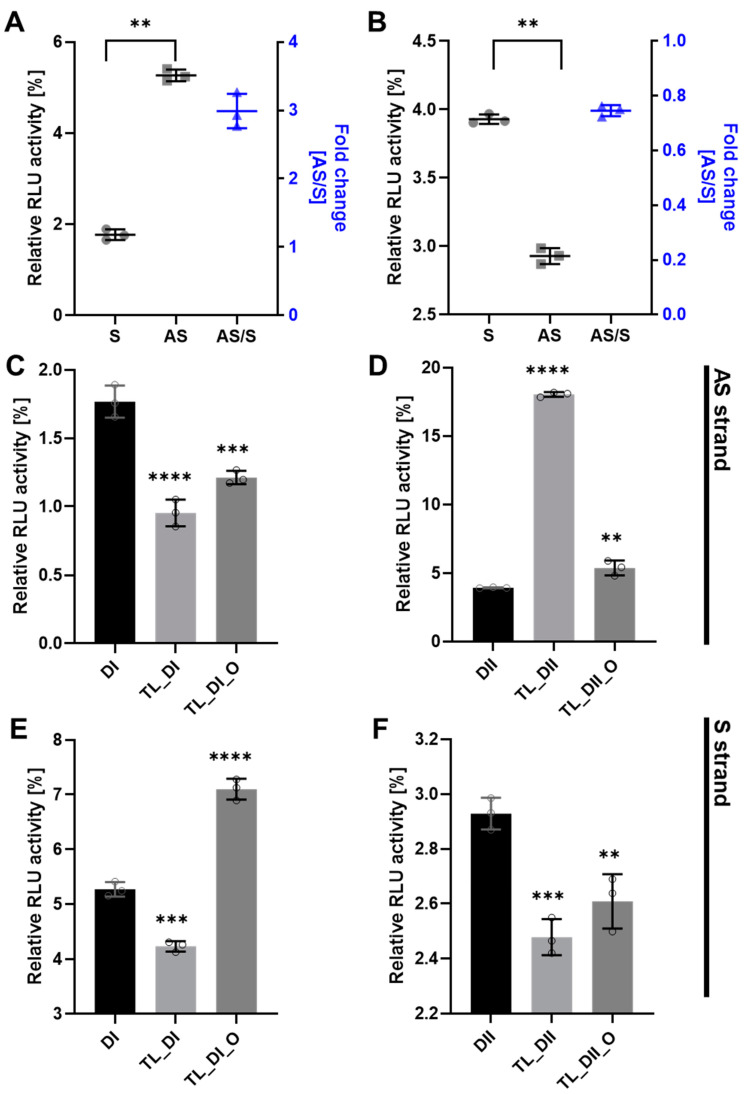
DsiRNA efficacy in HCT116 cells. Humanized Renilla luciferase was cloned in either the sense or antisense orientation into hnRNPH1 transcripts to act as a reporter. Relative expression of Renilla luciferase was determined by dual-luciferase assay against an internal control of firefly luciferase; Renilla activities were normalized to those of firefly and arbitrarily set at 100. Data represent the mean ± S.D. from three independent experiments (Student’s *t*-test ** *p* < 0.01, **** p* < 0.001, and ***** p* < 0.0001). We transfected the dual-luciferase reporter (S reporter targeting the sense strand of DsiRNAs, AS reporter targeting the antisense strand of DsiRNAs) and variant DsiRNAs into HCT116 cells. The DsiRNAs represented are: (**A**). DsiRNA I, (**B**). DsiRNA II, (**C**): DsiRNA I_S, (**D**): DsiRNA II_S, (**E**): DsiRNA I_AS, and (**F**): DsiRNA II_AS.

**Figure 7 genes-13-00436-f007:**
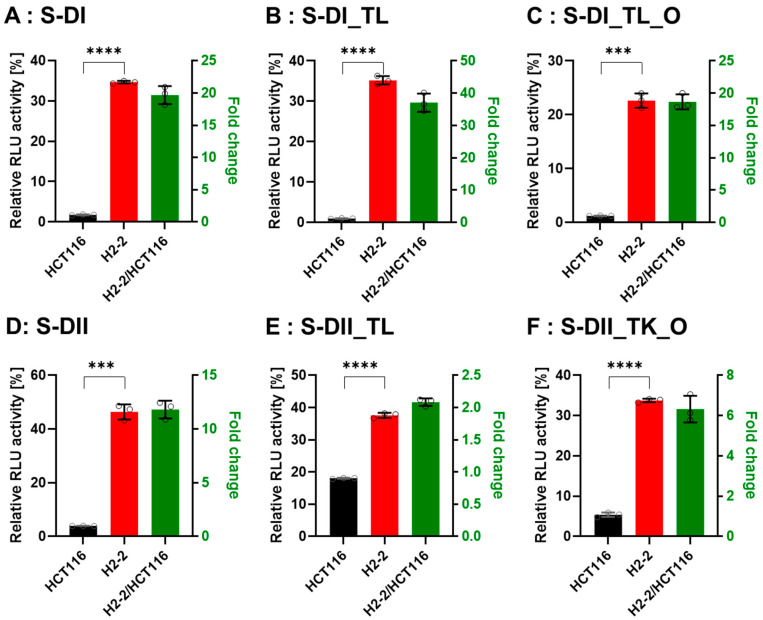
Comparison of DsiRNA efficacy in HCT116 and Dicer knockout H2-2 cells. The detection method used was the same as in Figure 6. We transfected the dual-luciferase reporter and DsiRNAs into HCT116 (black) and Dicer knockout H2-2 (red) cells. We determined the fold decrease of gene silencing efficiency in H2-2 by calculating the activity of H2-2 divided by HCT116 (green bar), as represented on the right y-axis. Graphs show the gene silencing activity of sense (S) strands (**A**–**F**) of (**A**): DI, (**B**): DI_TL, (**C**): DI_TL_O, (**D**): DII, (**E**): DII_TL, and (**F**): DII_TL_O, and antisense (AS) strands (**G**–**L**) of (**G**): DI, (**H**): DI_TL, (**I**): DI_TL_O, (**J**): DII, (**K**): DII_TL, and (**L**): DII_TL_O. Data represent the mean ± S.D. of three independent experiments (Student’s *t*-test ** p* < 0.05, *** p* < 0.01, **** p* < 0.001, and ***** p* < 0.0001).

**Table 1 genes-13-00436-t001:** The sequence of each strand of DsiRNAs. DsiRNA sequences were used to synthesize the tetra-looped DsiRNAs. n.t.; nucleotide.

Name	5′ to 3′	Size (n.t.)
S_DI	UGAAUCAGAAGAUGAAGUCAA	21
AS_TL_DI_O	AUdTdGGAAACAAUUUGACUUCAUCUUCUGAUUCAAG	35
AS_TL_DI	GCAGCCGAAAGGCUGCUUGACUUCAUCUUCUGAUUCAAG	38
S_TL_DII_O	GAAUCAGAAGAUGAAGUCAAAUUdGdGGAAACCAAU	35
S_TL_DII	GAAUCAGAAGAUGAAGUCAAGCAGCCGAAAGGCUGC	36
AS_DII	UUGACUUCAUCUUCUGAUUCAA	22

## Data Availability

Data sharing not applicable.

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
