# Peer review of "The Effect of Dicer Knockout on RNA Interference Using Various Dicer Substrate Small Interfering RNA (DsiRNA) Structures"

_genes, 2022, doi:10.3390/genes13030436_

Round 1
Reviewer 1 Report
RNA interference is gene silencing process in which targeted transcripts are selected for processing by complementary short RNAs (siRNAs). Targeting siRNAs are generated in this process from long double stranded RNA by RNAse III endonuclease Dicer, which digests long dsRNA into 21 to 23nt duplexes and cooperates in selecting the active strand of siRNA and its loading into Argonaut-type effector. The Dicer function can be bypassed, although with limited efficiency, by supplying chemically synthetized 21nt siRNA duplexes. More effective silencing is achieved by chemically synthetized duplexes mimicking the double stranded substrate of Dicer RNAse (DsiRNAs). In the submitted manuscript authors evaluated silencing activity of double stranded Dicer substates and showed that inclusion of tetra-loop in stem-loop design of DsiRNAs improves their gene silencing activity. Authors compared activity of linear and tetra looped DsiRNAs in wild type and Dicer defective cells and showed that Dicer processing is required for silencing activity of DsiRNAs.
The manuscript by Min-Sun Song et al demonstrates utility of tetra-looped DsiRNAs design in gene silencing applications and contributes to our understanding of Dicer substrate interaction. While in general the work is appropriate for Genes, I believe that it deserves additional advancement, possibly by investigating the optimal tetra-loop distance from the Dicer cut sites, work that would be of great input for RNAi silencing design. The text and data are well organized, the argumentation and language of the manuscript are clear.
Specific suggestions:
- A) The western blot analysis demonstrating Dicer depletion in CRISPR-treated cells is not very convincing. The Dicer depletion can be further demonstrated by monitoring derepression of Dicer targeted transcripts such as Alu RNA.
- B) Can optimalization of the distance between Dicer cleavage sites and tetra-loop further improve gene-silencing effectivity? Such optimalization of design would be of great value for RNAi silencing applications.
Reviewer 2 Report
Song and colleagues investigate the role of Dicer in enhanced silencing by Dicer-substrate siRNAs with and without a tetra-loop. While the manuscript contains interesting information, not all conclusions are substantiated by the data. The authors are asked to address the following observations:
General:
It was already known from previous studies, including those by the same research group, that DsiRNAs are more effective in silencing target genes (higher efficiency and prolonged effect) and that DsiRNAs are processed by Dicer. It was proposed that Dicer promotes strand-specific loading of the diced siRNA into RISC. Here, DsiRNAs are used in HCT116 Dicer knockout cells to show that this strongly reduces silencing efficiency. This seems obvious for Dicer-substrate siRNAs. The authors should better explain why this observation provides, as claimed, a deeper understanding of Dicer function in the processing of DsiRNAs.
The inclusion of tetra-loops, mimicking a shRNA format, with a nick break in the guide strand could provide new information on the optimal design of silencing molecules. This was tested only in combination with the DsiRNA design. The effect of the loop on silencing efficiency was minimal (less than 2-fold). The conclusion (lines 21-22 and 414-417) that the increased silencing efficiency provided by the loop is mediated by Dicer is not supported by the data (Figure 7). DsiRNAs without the tetra-loop showed that same Dicer-dependency. Thus, this observation could be fully explained by the Dicer-substrate design of the siRNAs. The authors should include regular siRNAs with and without tetra-loop in their experiments. By the way, without a regular siRNA control the expected Dicer-dependency of the Dicer-substrate design is also not formally shown.
The authors should clarify their nomenclature for active strand, with the sense and the antisense strands being the active strand in the two different DsiRNAs. I would think active = guide = antisense.
The manuscript needs editing. It is rather sloppy, with missing characters, brackets, etc. In particular the section lines 391-407 is very poor. Some parts are redundant, e.g. in lines 169-176 and lines 185-189.
Specific:
The y-axes in several Figures are misleading (6B, 6E, 6F, 7I, 7L). They should start off at 0 to prevent over-interpretation of differences.
The increased silencing efficiency by DsiRNAs is not reported consistently. In line 66, this is “up to 10-fold”, in lines 14 and 201 this is “up to 100-fold” and “up to 100 times”.
There is a double self-citation. References 3 and 19 are identical.
In the Figure 3 legend, the panel A and B description is lacking. Arrow heads in the figure are only shown for the control, while this seems more important for the test samples.
Figure 5A also shows a Dicer band reacting with Ab14601 in the H2-2 cell lysate, contrasting the statement on lines 269-272.
In Figure 5B it is not clear where the stop codon (in red) is located in the sequence.
In Supplementary Figure 2A, the figure shows clones 16 and 20 whereas the legend mentions clones 16 and 17.
In Figure 6 the indication of the Sense and Antisense strands at the right appears to be swapped.
What is the text on lines 351-353 about? This reads like an editorial instruction.
Reviewer 3 Report
The presented article deals with Dicer-substrate small interfering RNAs (DsiRNAs). Several DsiRNAs variants were designed to test a role of the loop sequence on their efficiency. Role of Dicer enzyme in their processing and activity was tested as well.
The Introduction and Methods sections are well written with all the relevant information present.
However, I have several concerns regarding Results and Discussion sections:
- HCT116 cell line use is a near diploid which does not mean that it contains 1 copy of each chromosome, as indicated in the text (line 206). Neither included citations prove single-copy chromosomes. It should be verified that (at least) the Dicer locus is present in a single copy as this could be a source of confusion (see points 3 and 4).
- Only 1 Dicer-KO clone was selected for further experiments, which lowers the conclusions made by this study. Authors obtained 3 independent clones all of them bearing mutation in the Dicer locus (at least according to Fig. 3). Why only 1 clone was selected (and verified by sequencing) is not clear. Anyway, performing experiments on additional clone(s) will strengthen the conclusions.
- The confirmation of Dicer-KO is confusing. On the RNA level, there are 9-nts missing which makes the mutated allele still in-frame and Dicer protein is probably still expressed – as is indicated by WB analysis, albeit again with a little confusion (1 out of 2 Abs were working). The presence of the STOP codon is indicated in Fig. 5B, however, I cannot see any such a codon (TAA, TAG, TGA). It seems to me that a simple 9-nt deletion and AAs changes are present in the mutated allele. Based on the presented results, I would expect Dicer to be still expressed.
- As for point 4, I do not believe that Dicer mRNA is completely missing in mutated H2-2 clone. There is no reason for this – expression should be verified by more primer pairs located in the different regions of Dicer mRNA. If nonsense-mediated decay is still considered as an option, presence of the STOP codon should be clearly indicated and confirmed (I cannot see it from the presented data).
- Conclusions made out of data seem to me exaggerated. For example, claimed strand selectivity is based on residual RLU activity 2% and 6%, respectively (Fig. 6A). Albeit mathematically the result is statistically significant, “real” effect is almost the same as ~5% residual activity is very good inhibitory response. Also claiming “did not change overall strand selectivity” (line 319) is not exact, at least for DsiRNAII (Fig. 6D, F) with nearly 10-times difference.
- Effect of Dicer KO is much lower than expected, especially for DsiRNAII, AS strand (changes from ~ 3% to ~ 6% inhibition; Fig. 7J,K,L) - see my comments in point 5. These results point to rather (or at least partial) Dicer-independent action. A possibility is mentioned in the main text (e.g. the role of nicks present in DsiRNAs, lines 325-330), however, this should be discussed more properly.
Minor points:
- Lines 185-196 and 238-245 seem to me rather as Figure legends (for Fig. 1 and 3, respectively) than as a part of the main text (as they are presented) – otherwise, the information is duplicated in the text.
- Figure 4A is better fitting to Fig. 2. Similarly, the corresponding main text (lines 248-250) is not fitting well into the paragraph.
- Label in Fig. 5D is not aligned properly.
- Describing of the luciferase reporter system is somehow confusing (lines 313-314 in the main text and lines 333-334 in the Fig. 6 legend compared to description in the Methods). It should be unified properly.
- Data presented in Fig. 6 and supplementary Fig. 3 seem to be identical. Personally, I prefer the format in supplementary Fig. 3 which seems to me more "familiar" and better to “read out” (at least because of the same scale on the Y axis for all samples to be compared).
- Lines 351-353 are probably not intended to be present in the main text (it seems to me as a comment from a colleague).
Overall, I would recommend to clarify the situation with Dicer KO, to prove that Dicer is really not present and expressed any more – for me, this is fundamental for the presented conclusions. I would also recommend to verify results on an independent Dicer-KO clone(s) and discuss in more details the lack of expected Dicer KO effect (complete inhibition of DsiRNAs action).
Round 2
Reviewer 1 Report
I would like to thank authors for considering my suggestions.
After reviewing the revised manuscript, I concluded that authors satisfactory addressed my concerns regarding the efficacy of Dicer depletion by additional experimental data included in Fig.S4.
However, I still consider the analysis presented in paper to be of insufficient utility for publication in Genes.
Reviewer 2 Report
While some of the specific issues are addressed and clear errors in the manuscript are corrected, I am disappointed by the author’s responses to the main points of critique. They choose to either not address the issue, or to avoid it by reproducing previously published work that is beside the point.
My main critiques remain, i.e. (i) that Dicer affects the activity of DsiRNAs was already known, hence their name; (ii) the effect of the loop on silencing efficiency was minimal; and (iii) the claim that the effect of the loop on silencing efficiency is mediated by Dicer is not substantiated by the data.
Reviewer 3 Report
The revised version of the presented article is now improved. The main weakness, an insufficient proof of Dicer knock-out in the cell line used, has been resolved. Sufficient data is now presented to support a real lack of Dicer expression/activity in H2-2 clone thus making the conclusions more robust.
However, some minor points are still remaining that should be corrected, at least in my opinion:
- The presence of a stop codon, which was not illustrated and is not present in the H2-2 clone, is still mentioned in lines 272 and 273. I would recommend to delete it or reformulated without mentioning stop codon presence, because the evidence is missing.
- A paragraph (lines 233-239, formerly 238-245) is still insufficiently corrected in my opinion and the text is misleading. I would recommend to cite the Fig. 3 appropriately, the best in line 236: … gel electrophoresis (Figure. 3). Also panels A and B (line 239) should be cited as (Figure 3A) and (Figure 3B), current version seems to me to be suitable for figure legend rather than main text.
- A sentence on lines 243-245 is not fitting into the text which is dealing with PCR and sequencing and information provided in this sentence is over-abundant and better suits into a section dealing with CRISPR knock-out.
- The 2 new paragraphs added in Discussion section contain a little bit of orverlapping information and some methodological details are described too in-depth for discussion section (could be mentioned in Methods instead). However, I would also add a section dealing with Dicer-independent effects, which are still not sufficiently discussed and are important for proper data interpretation. A comment on my “point 6”: “After Dicer KO, all DsiRNAs showed good gene-silencing activity. These results support that there exist noncanonical pathways for DsiRNA biogenesis, which bypass a part of the biogenesis steps of Dicer cleavage. As for Dicer-independent biogenesis, mature miR-451 can be produced without the Dicer cleavage step. Pre-miR-451 is cleaved by Ago2 in the middle of the 3' strand, and further trimmed by 3’-5' exoribonuclease PARN to yield a mature form of miR-451(2). DsiRNAs may have alternative pathways like pre-miR-451 that do not require a Dicer cleavage step during its biogenesis.” seems to me very clear and focused and definitely deserves to be included into the main text, Discussion section.
Author Response
Please see the attachment
2nd Respond reviewer 3
The revised version of the presented article is now improved. The main weakness, an insufficient proof of Dicer knockout in the cell line used, has been resolved. Sufficient data is now presented to support a real lack of Dicer expression/activity in H2-2 clone thus making the conclusions more robust.
However, some minor points are still remaining that should be corrected, at least in my opinion:
- The presence of a stop codon, which was not illustrated and is not present in the H2-2 clone, is still mentioned in lines 272 and 273. I would recommend to delete it or reformulated without mentioning stop codon presence, because the evidence is missing.
We would like to thank the reviewer for the positive feedback. We clarified the description on figure 5 as follow:
Results section in lines 269-288
"To confirm that the Dicer was knocked out in H2-2 cells, we compared the amino acid se-quence in gDNA isolated from HCT116 and H2-2 cells (Figure 5B). Figure 5C showed the Dicer mRNA sequence targeted with dicer gRNA. The H2-2 Dicer knockout cells exhibited a deletion of three amino acids and a mutation of seven amino acids. The 9-nt missing still in-frame in Dicer proteins and the potential AUG initiation codons in the H2-2 genomic DNA sequencing can generate open reading frames that overlap the Dicer reading frame (Supplementary Figure 2). To overcome the potential shortcomings of this approach, which include for example initiation of translation from an alternative Dicer expression that is hard to quantify, we analyzed the impact of Dicer expression via Mass spectrome-try. Dicer unique peptides only showed in the sample from HCT116, which generated the gel band extraction from closed to Dicer protein size in supplementary table 1. We used the TaqManTM MicroRNA assay to establish the temporal dynamics of miRNA depletion following induction of Dicer loss of function (Supplementary Figure 4). miR-1254 is a non-canonical miRNA produced from an intron of protein-coding gene, CCAR1. The posi-tion of pre-miR-1254 overlaps with that of the Alu sequence, a type of short interspersed nuclear elements (SINEs), and belongs to the Alu Jr subgroup. miR-1254 is dependent strictly on Dicer [31]. miR31 is canonical miRNAs. Our analysis of the abundant miR-31 and miR-1254 in HCT116 and H2-2 confirmed that the levels of the abundant miRNAs in Dicer knockout cells are at least 90%(miR-31) and 70%(miR-2154) lower than in wild-type cells(Supplementary Figure 4)."
- A paragraph (lines 233-239, formerly 238-245) is still insufficiently corrected in my opinion and the text is misleading. I would recommend to cite the Fig. 3 appropriately, the best in line 236: … gel electrophoresis (Figure. 3). Also panels A and B (line 239) should be cited as (Figure 3A) and (Figure 3B), current version seems to me to be suitable for figure legend rather than main text.
We would like to thank the reviewer for bringing our attention to the necessity for revising main text. We revised this point in 3.2. Generation of Dicer knockout HCT116 cells.
Results section in lines 236-241
"Positive control used the control C/G that a 633 bp control DNA with a point mutation (Supplementary Figure 1B) demonstrates the expected 416 bp and 217 bp bands. Homodu-plex DNA without mismatch did not cleave the nuclease, but heteroduplex DNA (Dicer H2-1, H2-2, and H2-3) show a cleavage band in generating site-specific double-strand break (Figure 3). Hereafter, all clones tested were efficacious in facilitating the cleavage of DNA at specific targets in the Dicer genome."
- A sentence on lines 243-245 is not fitting into the text which is dealing with PCR and sequencing and information provided in this sentence is over-abundant and better suits into a section dealing with CRISPR knockout.
Thank you so much for your comment. We agree with Reviewer's comment. Due to the remarkably rapid technological advancement, many CRISPR techniques are now published. We feel it is our duty to inform our readers which of the many CISPRs we have used to create Dicer knockout cells because the generation of Dicer knockout cells is a critical point in the manuscript.
- The 2 new paragraphs added in Discussion section contain a little bit of orverlapping information and some methodological details are described too in-depth for discussion section (could be mentioned in Methods instead). However, I would also add a section dealing with Dicer-independent effects, which are still not sufficiently discussed and are important for proper data interpretation. A comment on my "point 6": "After Dicer KO, all DsiRNAs showed good gene-silencing activity. These results support that there exist noncanonical pathways for DsiRNA biogenesis, which bypass a part of the biogenesis steps of Dicer cleavage. As for Dicer-independent biogenesis, mature miR-451 can be produced without the Dicer cleavage step. Pre-miR-451 is cleaved by Ago2 in the middle of the 3' strand, and further trimmed by 3’-5' exoribonuclease PARN to yield a mature form of miR-451(2). DsiRNAs may have alternative pathways like pre-miR-451 that do not require a Dicer cleavage step during its biogenesis."seems to me very clear and focused and definitely deserves to be included into the main text, Discussion section.
We are grateful for this comment. We revised our manuscript to present and discuss this information in the Results and Discussion parts.
Results section in lines 354-361
"After Dicer knockout, all DsiRNAs showed good gene-silencing activity (Figure 7). These results support that there exist noncanonical pathways for DsiRNA biogenesis, which by-pass a part of the biogenesis steps of Dicer cleavage. As for Dicer-independent biogenesis, mature miR-451 can be produced without the Dicer cleavage step. Pre-miR-451 is cleaved by Ago2 in the middle of the 3' strand, and further trimmed by 3’-5' exoribonuclease PARN to yield a mature form of miR-451[63]. DsiRNAs may have alternative pathways like pre-miR-451 that do not require a Dicer cleavage step during its biogenesis."
Discussion section in lines 399-424
"As far as Dicer knockout HCT116 cells are concerned, although Dicer helicase mutant HCT116 cells have been used to validate the efficiency of various shRNAs in our previous studies [64]. In previous studies, Dicer helicase mutant HCT116 did not affect the mature short RNA generation of miR-23a, miR-27a, and artificial shRNAs. This has proved to be very difficult because of Dicer helicase mutant HCT116 with defects in the processing of most, but not all, endogenous pre-miRNAs into mature miRNA. Another college group showed that a stem-looped miRNA does not require a Dicer cleavage step during its biogenesis [63,65]. The biogenesis of miR-451 occurs independently of Dicer and instead requires cleavage of the 3' arm of the pre-miR-451 precursor hairpin by Ago2. This is evidence that not all stem-looped RNAs are affected by their efficiency by Dicer. Our studies demonstrated how we improved the knockout cells and knockout validation by the CRISPR system. To get the Dicer knockout cell lines, we are considered three things. First, we chose the HCT116 cell line because it is near diploid and often used for gene knockout studies [31, 32]. The diploid cell lines offer a complete loss-of-function phenotype from a single allele knockout and eliminate any activity of the knockout from a second allele seen in diploid cell models [47-49]. Second, we collected a single cell harboring Dicer knockout after transfection of CRISPR plasmid selected GFP-only expression cells by FACS sorting. The method can generate a new cell line that can be verified as complete knockouts [50-52]. Finally, we used two methods of western blot and Dicer mRNA from qRT-PCR to confirm the knockout validation. The Dicer antibody of Ab13502 detected the band from lysates of H2-2, which was Dicer knockout cells. However, the Dicer Ab14601 antibody exhibited target specificity and sensitivity to allow the identification of Dicer protein (Figure 5A). It is of utmost importance to find an excellent antibody to reduce the cross-reactivities with off-target proteins that can lead to the recognized issue of experimental irreproducibility [53, 54]. We clearly confirmed the knockout validation from the sequences detected of Dicer mRNA from HCT116 and H2-2 (Figure 5D and Supplementary Figure 3)."
We are hopeful that this amended version is now suitable for publication in the Genes Journal.
